# Antitumor and Antibacterial Activity of Ni(II), Cu(II), Ag(I), and Hg(II) Complexes with Ligand Derived from Thiosemicarbazones: Characterization and Theoretical Studies

**DOI:** 10.3390/molecules28062590

**Published:** 2023-03-13

**Authors:** Heba Alshater, Ahlam I. Al-Sulami, Samar A. Aly, Ehab M. Abdalla, Mohamed A. Sakr, Safaa S. Hassan

**Affiliations:** 1Department of Forensic Medicine and Clinical Toxicology University Hospital, Menoufia University, Shebin El-Kom 32511, Egypt; 2Department of Chemistry, College of Science, University of Jeddah, Jeddah 21589, Saudi Arabia; 3Department of Environmental Biotechnology, Genetic Engineering and Biotechnology Research Institute, University of Sadat City, Sadat City 32958, Egypt; samar.mostafa@gebri.usc.edu.eg; 4Chemistry Department, Faculty of Science, New Valley University, Alkharga 72511, Egypt; ehababdalla99@sci.nvu.edu.eg; 5Medical Microbiology and Immunology Department, Faculty of Medicine, Suez University, Suez 41522, Egypt; 6Department of Chemistry, Faculty of Science, Cairo University, Giza 12613, Egypt; hsafaa@sci.cu.edu.eg

**Keywords:** complexes, DFT, antitumor, antibacterial, molecular docking

## Abstract

Four new complexes (Ni^2+^, Cu^2+^, Ag^+^, and Hg^2+^) were prepared from the ligand N-(4-chlorophenyl)-2-(phenylglycyl)hydrazine-1-carbothioamide (H_2_L). Analytical and spectroscopic techniques were used to clarify the structural composition of the new chelates. In addition, all chelates were tested against bacterial strains and the HepG2 cell line to determine their antiseptic and carcinogenic properties. The Ni(II) complex was preferable to the other chelates. Molecular optimization revealed that H_2_L had the highest reactivity, followed by Hg-chelate, Ag-chelate, Ni-chelate, and Cu-chelate. Moreover, molecular docking was investigated against two different proteins: the ribosyltransferase enzyme (code: 3GEY) and the EGFR tyrosine kinase receptor (code: 1m17).

## 1. Introduction

For decades, thiosemicarbazone (TSC) derivatives have received much attention and research interest; they are a type of inorganic metal chelator that serve as simple ligands to form complexes with a variety of transition metals, such as Ni, Cu, Ag, Hg, and others [1,2]. TSCs and similar complexes offer important chemical properties due to their diverse connection mechanisms, advantageous organic inferences, organizational diversity, and ion-detection capabilities [3]. They have also been utilized in therapeutics and have a wide range of organic properties, including antiseptic [4], antifungal [5,6], antiamoebic [7], antiviral [8], antitumor [9], and antimalarial [10] activities. Furthermore, numerous publications have demonstrated the biological characteristics of metal complexes containing nitrogen–sulphur, oxygen–nitrogen–sulphur, and oxygen–nitrogen–nitrogen chelating derivatives employing several carbonyl compounds [11]. Some of the literature has reported the biological activity of thiosemicarbazone derivatives and complexes of Ni^2+^, Cu^2+^, Ag^+^, and Hg^2+^. The antimicrobial activity against Gram-positive and Gram-negative bacteria and fungi was tested for three nickel(II) thiosemicarbazone (L_1_H, L_2_H, and L_3_H) complexes, where L_1_H is (E)-4-(4-fluorophenyl)-1-(1-hydroxypropan-2-ylidene) thiosemicarbazide, L_2_H is (E)-4-(4-chlorophenyl)-1-(1-hydroxypropan-2-ylidene) thiosemicarbazide, and L_3_H is (E)-4-(4-chlorophenyl)-1-(1-hydroxypropan-2-ylidene) thiosemicarbazide; the antituberculosis activity was also tested on all compounds. These findings demonstrated the potential of Ni^2+^ chelates for developing antibacterial and antitubercular chemotherapeutic drugs [12]. The antimicrobial activity of the ligands N-(4-Methoxyphenyl)-2-(5-(morpholinosulfonyl)-2-oxoindolin-3-ylidene)hydrazine-1-carbothioamide and their Cu (II) and Zn (II) complexes were evaluated against two Gram-positive strains; both complexes revealed antimicrobial activity against the tested strains [13]. The antibacterial effect of the Ag(I) complex [Ag(catsc)(PPh_3_)_2_]NO_3_ (catsc = 3-phenylpropenalthiosemicarbazone) was premeditated against the normal straining of two Gram-positive (*Enterococcus faecalis* and *Staphylococcus aureus*) and two Gram-negative (*Pseudomonas aeruginosa* and *Escherichia coli*) microbes. The antibacterial activity of the Ag(I) complex is enhanced concerning the free ligand [14]. Additionally, the antimicrobial properties of Hg(II) complex with ligand 2-formyl pyridine thiosemicarbazones were assessed against several microorganisms. According to the screening results, the Hg(II) metal chelates have a more potent inhibitory impact than the parent ligand [15]. Herein, we reported the preparation and characterization of Ni^2+^, Cu^2+^, Ag^+^, and Hg^2+^ chelates with ligand N-(4-chlorophenyl)-2-(phenylglycyl)hydrazine-1-carbothioamide. Ligand and its complexes were assessed against the HepG2 human liver cancer cell line and some bacterial strains.

## 2. Results

### 2.1. Physicochemical Properties

The results presented that complexes 1S–4S are colored, stable in air, and insoluble in most organic solvents except DMF or DMSO. The elemental analysis and some physical properties are exposed in Table 1. Elemental analyses indicated that the complexes 1S and 4S are moulded in a 1:1 (M:L) molar ratio, while 2S and 3S are formed in 1:2 and 2:1 (M:L), respectively, which matched with the proposed formula. The molar conductivity in 10^−3^ M DMF solution indicates that they are non-electrolytes by nature [1,2,3].

### 2.2. FT-IR

The FT-IR spectra of ligand and its metal complexes are depicted in Table 2 and Figure 1 and Appendix A. The FT-IR spectrum of the ligand shows stretching frequencies of ν(N4), ν(N2), ν(N1), ν(C=O), and ν(C=S) at 3335, 3302, 3100, 1670, and 750 cm^−1^, respectively [3]. On complexation, these bands were shifted to lower and higher frequencies (3641–3296, 3294–3102, 3018–2924, 1674–1627, and 756–755 cm^−1^) for Ni(II), Cu(II), Ag(I), and Hg(II) complexes, respectively. The red shift proves that the azomethine nitrogen atoms share in complex forming. The appearance of new bands within the ranges 462–416 (νM–N) [4,5,6] and 549–501 (νM–O) cm^−1^ [7,8,9] confirms the participation of the N atom of the azomethine group [10] and the (carbonyl) O atom in the formation of the complexes to form a hexagonal ring with a carbaldehyde moiety. On the other hand, the broad bands at 956 and 946 were assigned to the coordinated OH and H_2_O for Ag(I) and Hg(II) complexes, respectively. Additionally, the bands at 3422 and 3471 were assigned to hydrated water in Ni(II) and Hg(II) complexes.

The presence of bands at 1634–1595 cm^−1^ was assigned to ν(CO_3_), indicating bidentate carbonate groups which also correspond to O–C–O asymmetric and symmetric stretching vibrations [11,12,13].

### 2.3. ESI-MS Spectra

The mass spectrum of the ligand exhibited a molecular ion peak at *m*/*z* = 336 amu (Calc. *m*/*z* = 334.5). The important fragment ions appear at: *m*/*z* 77 for [C_6_H_4_-3H]^+^, 112 for [C_6_H_5_Cl-H]^+^, 127 for [C_6_H_5_NCl-2H]^+^, 201 for [C_7_H_9_N_3_SCl]^+^, 228.5 for [C_8_H_9_N_3_SOCl-H]^+^, 259 for [C_9_H_10_N_4_OSCl-2H]^+^, and 336 for [C_15_H_15_N_4_OSCl] (Appendix A).

The mass spectra of Ni(II), Cu(II), Ag(I), and Hg(II) complexes shows molecular ion peaks at *m*/*z* 472, 732, 618, and 717; these data are in good agreement with the proposed molecular formulas for complexes (calc. 471.54, 732.18, 618.65, and 715.35 amu), respectively (Appendix A).

The interpretation of the mass spectra of the complexes Cu(II) and Ag(I) has been clarified, revealing several significant fragments. The data are in a good agreement with the proposed molecular formulas for complexes Cu(II) and Ag(I) (calc. 732.18 and 618.65, respectively), and this verifies the complexes’ chemical structures.

The mass spectra of the Cu(II) complex shows the peak attributed to the molecular ion peak [M+] *m*/*z* at 732 (8.52%) corresponding to C_30_H_29_C_l2_CuN_8_O_2_S_2_. The various fragments of this complex provide the peaks with different intensities at different *m*/*z* positions at 348.91 (100.00%) (C_15_H_15_ClCuN_3_OS), 620.71 (11.20%) (C_25_H_27_ClCuN_7_O_2_S_2_), 585.03 (19.20%) (C_22_H_22_C_l2_CuN_6_OS_2_), 578.58 (22.59%) (C_24_H_29_ClCuN_7_O_2_S), and 91.12 (17.94%) (C_6_H_5_N).

The mass spectra of the Ag(I) complex displays a peak recognizable to the molecular ion peak [M+] *m*/*z* at 618 (9.22%) corresponding to C_15_H_19_Ag_2_ClN_4_O_3_S2. The various fragments of this complex provide the peaks with different intensities at different *m*/*z* positions as at 127.57 (100.00%) (C_6_H_6_ClN), 600.64 (13.20%) (C_15_H_17_Ag_2_ClN_4_O_2_S_2_), 347.19 (9.20%) (C_10_H_15_AgN_4_OS), 90.15 (14.59%) (C_2_H_6_N_2_S), and 75.09 (8.94%) (C_2_H_7_N_2_O).

### 2.4. Electronic Spectral Bands

The electronic spectral bands of the H_2_L and Ni^2+^, Cu^2+^, Ag^+^, and Hg^2+^ complexes (λmax, nm) in the DMF solution were scanned in the range of 190–810 nm at room temperature. The values of the λmax and magnetic moments (μeff) are recorded in Table 3. The UV spectra of ligand were observed as two absorption bands at 260 and 300 nm assigned to the π–π* transition [3]. On complexations, the UV-visible spectra of Ni(II), Cu(II), Ag(I), and Hg(II) complexes were shifted to higher wavelength exhibit bands at 286 and 374, 281 and 371, 281, and 299 and 378 nm, respectively, which may be assigned to π -π*, n-π* transitions, and showed no d-d band, representing a square planar geometry for Cu^2+^, Ag^+^, and Hg^2+^ complexes, although Ag compounds were known mostly as two-coordinated, but the square planar silver(I) complexes increasingly observed stereochemistry for silver(I), as was found in our synthesized complex [14,15,16]. In contrast, the Ni(II) complex is square pyramidal geometry. The difference in the λ max values of the H_2_L and its complexes can approve the coordination of the ligand with metal ions [17,18].

### 2.5. PXRD of Ligand and Metal Complexes

The X-ray diffract gram patterns of ligand H_2_L and its complexes of Ag(I) and Hg(II) complexes were evaluated in Table 4 and Figure 2, Figure 3 and Appendix A. The powder diffraction patterns were recorded over the (2θ = 5–90) range lattice constants. The intensities of the powder lines and the corresponding 2θ values are found to be different between ligand and complexes, indicating their crystalline nature. The average particle size of the crystalline ligand (H_2_L) and its complexes was calculated using Scherer’s equation [19,20,21]. The Scherer’s constant (K) in the formula provides for the particle’s shape and is commonly assumed to be 0.9. It was found that the calculated crystalline size was in the nano range. The values of the crystallite size for H_2_L, 3S, and 4S are 41.50, 35.12, and 41.02, 54.35, 52.03, and 26.75, and 52.28, 50.11, and 64.12 nm, respectively.

### 2.6. Thermal Analysis

TGA plays a crucial role in assessing the features of new metal complexes, discovering the different solvent molecules inside or outside the coordination sphere, and figuring out the thermal stability of the complexes. In addition to the microanalyses’ outcomes, the TGA results, which were carried out between 20 and 800 °C, were utilized to assess and compute mass loss. TGA analyses were carried out for ligand and Ni^2+^(1S), Cu^2+^(2S), Ag^+^(3S), and Hg^2+^(4S) complexes (Table 5). At temperatures between 190 and 633 °C (Calc. 100 %, found 99.9 %), the ligand TGA plot shows their complete thermal breakdown in one step.

Three weight loss events were visible in the TG curves of the Ni(II) complex. The first decomposition step occurred between 41 and 178 °C and was accompanied by weight losses of 3.82 (3.86): Calc./ found %, which are interpreted as losses of the hydrated water molecule. The second phase occured between 178 and 288 °C and involved the losses of the C_15_H_14_N_4_OS and moiety with weight losses from the complexes estimated to be 63.27 (63.21) Calc./ found %. In the third phase, the temperature was between 288 and 391 °C with weight losses of 17.06 (17.11) Calc./found % corresponding to HCl and CO_2_ molecules losses, and NiO remaining as the final residue. While in the Cu(II), Ag(I), and Hg(II) complexes, the TGA plot showed the complete decomposition of organic molecules in one step between a temperature range of 105 and 385 °C with weight losses of 91.32, 56.72, and 59.65 (91.28, 56.76, 59.60) Calc./ found % and the remainder of Cu, 2AgO + 3C, and HgO + 6C as the final residues, respectively.

### 2.7. DFT Calculations of the Ligand and Metal Complexes

Table 6 revealed the geometric information of the H_2_L and its metal chelates as the energy, dipole moment, hardness, softness, chemical potential, and electronegativity. Researchers were investigating the frontier molecular orbitals LUMO (p acceptor) and HOMO (p donor) that represents the charge transfer interface inside the LUMO–HOMO molecule leading to the appearance of the parameters, the hardness and softness, which are commonly used as a criterion of chemical reactivity and stability (Figure 4). The smaller hardness values imply a higher reactivity, which means that a molecule with a small HOMO-LUMO gap is more reactive and softer. The hardness and softness can be calculated using the equations η(hardness) = (I − A)/2; S(softness) = 1/2η. The compounds’ reactivity is arranged as follows: H_2_L > Hg-chelate > Ag-chelate > Ni-chelate > Cu-chelate. The energy gap for all the complexes is higher than that of the ligand. So, the investigated complexes are more stable than the parent ligand; accordingly, the Cu-chelate was the most stable. The geometrical molecular structure provided the atomic ordering of H_2_L, and complexes are also included (Figure 5). After investigating the computed bond lengths and orientations of ligand and its metal chelates, we observed some changes after coordination, as presented in Appendix A. Furthermore, many bond lengths were elongated as N12-N11, C9-O10, C8-N7, and C13-N12 to adjust for the coordination via the N12 and O10 in all chelates. The thione group shared in the coordination in both 1S and 3S chelates with the formation of new M-S bonds.

In the case of Ni, Ag, and Hg chelates, every metal completed its coordination with other chelating agents and the parent ligand, such as carbonate, water, and iodide and hydroxide. Table 6 illustrates the bond lengths of newly constructed ligand bonds. Additionally, new bond angles were observed, and others were changed to optimize the coordination as C16-N15-C13, N15-C13-Nl2, C13-N12-Nl1, and O10-C9-N11. The negative charge is delocalized over N12 and O10 with calculated charges of −0.473 and −0.459, respectively. Therefore, both are common donation sites in all chelates. After chelation, these charges were decreased, and the electron density above the metal ions increased due to the charge transfer from ligand to metal. The charges of N12 and O10 were changed to −0.395 and −0.345, −0.458 and −0.479, −0.391 and −0.300, and −0.465 and −0.362 in 1S, 2S, 3S, and 4S, respectively. It was noticed that some electron densities were increased due to the back donation from M to the ligand. The metal charges changed to +0.336, +0.456, +0.129, and +0.519 for 1S, 2S, 3S, and 4S, respectively.

### 2.8. Biological Applications

#### 2.8.1. Antibacterial Activity

The antibacterial activities of ligand H_2_L and Ni^2+^, Cu^2+^, Ag^+^, and Hg^2+^ complexes were screened against bacterial species, Gram-negative bacteria (*Escherichia coli* and *Klebsiella pneumonia*), and Gram-positive bacteria (*Staphylococcus aureus* and *Streptococcus mutants*). Ampicillin and gentamicin were used as the standards for antibacterial studies. The results of the antibacterial activity of the ligand and their complexes are presented in Appendix A and Figure 6. These results suggested that the complexes are more potent antibacterial agents than ligands due to their chelation ability. Whether ampicillin and gentamicin were used as standard drugs, the nickel complex in our study had the best antibacterial activity against bacterial species. The nickel complex showed more antimicrobial effects against *Escherichia coli* and *Klebsiella pneumonia* than gentamicin. This agrees with other researchers using nickel in their studies [21,22]. Similarly, it showed more antimicrobial effects against *Staphylococcus aureus and Streptococcus mutans* than ampicillin. This is in line with other similar research results [23]. Moreover, the mercury complex has a high antimicrobial effect against *Staphylococcus aureus* and *Streptococcus mutans* with an inhibition zone of 23.3 ± 0.6 and 44.3 ± 0.6, respectively, and is higher than that of the reference antibiotic. Furthermore, ampicillin has inhibition zones of 22 ± 0.1 and 30 ± 0.5 when testing *Staphylococcus aureus* and *Streptococcus mutans*, respectively. This is consistent with other studies that have used mercury [24,25]. Consequently, the resulting data showed that the antibacterial efficacy of new complexes against the Gram-negative bacteria could be arranged in the following order: Ni complex > gentamicin > Hg complex > Ag complex > Cu complex > Ligand. However, for *Staphylococcus aureus*, the sequence of antibacterial action was Ni complex > Hg complex > ampicillin > Cu complex > Ag complex > ligand. The same pattern was observed against *Streptococcus mutans*, except that the ligand has no antimicrobial activity. The effectiveness of synthesized compounds may appear to be lipophilic, which may block or impede the viable development of Gram-negative and Gram-positive bacteria by facilitating the diffusion of the compounds through the lipid bilayer membrane. This is explained by the chelation theory, which argues that a decrease in the polarizability of metal complexes or an increase in hydrogen bonding can increase the lipophilic susceptibility of the complex, hence enhancing its antimicrobial activity [26].

#### 2.8.2. Cytotoxicity

The MTT test was used to assess the in vitro cytotoxicity of the ligand H_2_L and its complexes with Ni(II), Cu(II), Ag(I), and Hg(II) against the human HepG2 cell line. In terms of optical thickness, the movement of mitochondrial dehydrogenase was determined to be an indication of cell viability. Non-linear regression methods were used to determine the IC_50_ values for the investigated substances during the experiment. The results are reported as the IC_50_, which is the concentration of a chemotherapeutic agent that produces a 50% reduction in cancer cell proliferation compared to control cell growth [27]. The cytotoxicity for the ligand was performed at concentrations of 3.125, 6.25, 12.5, 25, 50, and 100 μg/mL. At the same time, the cytotoxicity for the ligand complexes with Ni, Ag, Hg, and Cu was at concentrations of 31.25, 62.5, 125, 250, 500, and 1000 μg/mL based on the surviving fraction results and IC_50_ values (Appendix A and Figure 7). Among the investigated complexes, Ni(II) complex (IC_50_ = 41.2 µM) had the highest activity against the human HepG2 cell line. The IC_50_ values followed the this order: Vinblastine (4.58) < Ligand (20.45) < Ni(II) complex (41.2) < Hg(II) complex (48.5) < Ag(II) complex (293.95) < Cu(II) complex (182.61) µM. Moreover, the findings showed that the ligand and its complexes are potent against human HepG2 cell lines. Importantly, the Ni(II) complex was the most effective and exhibited concentration-dependent effects, suggesting its potential utility in cancer treatment.

#### 2.8.3. Molecular Docking Studies

Simulation of the protein–drug interaction is important in designing the structure-based drug [28,29]. Thus, we investigated the theoretical interaction between the prepared compounds with some proteins selected from the protein data bank. The ribosyltransferase enzyme (code: 3GEY) was selected for the antibacterial study, and the EGFR tyrosine kinase receptor (code: 1m17) was selected for the anticancer study. The interaction profile of the tested compounds with 3GEY and 1m17 is presented in Appendix A and Figure 8 and Figure 9. The antibacterial docking results revealed the potency of the examined compounds, as indicated by the negative values of the scoring energies or the different types of interactions, as presented in Figure 8. There are several interactions, most notably the side chain acceptor in all chelates and the arene cation interaction type, prominent in numerous compounds such as H_2_L, 1S, and 2S. Appendix A reflects the interactions for each compound.

H_2_L ligand: the backbone donor interaction between Leu-A624 and the oxygen amide group of the ligand. Additionally, arene cation is an interaction between Lys-A518 and the phenyl group of the parent ligand. Ni-chelate: backbone donor interaction between Asn-A553 and the carbonyl oxygen of the carbonate group, sidechain acceptor between Gln-A521 and the H atom of the NH group, and arene–arene between Phe-A642 and phenyl ring. Cu-chelate: arene–cation interaction between Lys-A525 and the phenyl ring, sidechain acceptor between Asp-A623 and Asn-B508, the H atom of the NH group, and a sidechain donor between the Gln-B549 oxygen amide group. Ag-chelate: arene–cation interaction between Lys-A518 and two of the phenyl rings and sidechain acceptor between Asn-B508 and (H) of the water molecule. Hg-chelate: sidechain acceptor between Asn-B508 and (H) of the NH group. Despite the fact that Cu-chelate has the highest scoring energy, experimentally, Ni-chelate had the highest antibacterial activity. It may be due to the various interactions observed only in Ni-chelate as the sharing of the carbonate group with a backbone donor interaction and the aromatic–aromatic interaction via the phenyl ring.

The investigation of the EGFR tyrosine kinase receptor (code: 1m17) explored the high affinity of the prepared compounds to the 1m17 protein. Hg-chelate had the highest significant negative scoring energy value that is compatible with the IC_50_ of chelates. The different types of interactions are represented in Figure 9 and Appendix A. The chelates show side chain donor (arene–cation and sidechain acceptor) and metal contact interactions for Ni-, Cu-, and Hg-chelates.

Finally, all bond lengths for most of the interactions were found to be less than 3.5 Å, the reported range for an actual docking track [30]. The obtained data indicated that the examined compounds are promising and recommend further bioactivity investigations.

## 3. Experimental Section

### 3.1. Material and Methods

The preparation tools and methods utilized for the structure confirmation and application of metal salts NiCO_3_, Cu(ClO_4_)_2_, Ag_2_S, and HgI_2_ were obtained as shown in the Appendix A.

### 3.2. Preparation of Ligand and Metal Complexes

The organic ligand N-(4-chlorophenyl)-2-(phenylglycyl)hydrazine-1-carbothioamide was synthesized and characterized according to Aly and Eldourghamy [21] and Abdalla et al. [3]. Ni^2+^, Cu^2+^, Ag^+^, and Hg^2+^ chelates were prepared by adding a stoichiometrically amount of target metal ions in absolute ethanol to a hot ligand solution in a 1:1 molar ratio as represented in Figure 1**.** The metal complexes were magnetically stirred at 60 °C for 5–9 h. The hot precipitous were filtered, leaving the solution at 35 °C to vaporize some solvents and promote the crystallization. The crystals were collected by vacuum filtrations, washed several times with anhydrous diethyl ether, and dried under a vacuum in the presence of phosphorus pentoxide (P_4_O_10_).

### 3.3. *Computational Study*

GaussView 5.0.8 (Wallingford, CT, USA, 2009) software was used to prepare the input files of compounds, as shown in Section 2 [31].

### 3.4. Antibacterial Assay

The antimicrobial activity of the prepared ligands and complexes(1S–4S) was conducted using the agar well diffusion method [32,33]. The H_2_L and complexes (1S–4S) were confirmed in vitro for their antibacterial activity against *staphylococcus aureus (ATCC:13565)* and *Streptococcus mutans* (ATCC:25175) (Gram-positive bacteria), *Escherichia coli* (ATCC:10536) and *klebsiella pneumonia* (ATCC:10031) (Gram-negative bacteria). All strains were stored at −80 °C before culture preparation. The strains were cultured in a suitable culture media; *Staphylococcus aureus* and *Streptococcus mutans* were cultured in blood agar, while *Escherichia coli* and *klebsiella pneumonia* were cultured in McConkey agar. Details of the antimicrobial screening methodology are illustrated in the Appendix A [34].

### 3.5. Cytotoxicity Assays

The MTT assay [3(4,5-dimethylthiazol-2-yl)-2,5-diphenyltetrazolium bromide] is a colorimetric examination for measuring the cell metabolic activity and proliferation [35]. In the current research, the MTT assay was used to test the cytotoxicity of the prepared ligand and its chelates against the HepG2 cell line. More details of the antitumor screening methodology are illustrated in the Appendix A [36,37,38].

## 4. Conclusions

The characterization and biological activity of four new (Ni^2+^, Cu^2+^, Ag^+^ and Hg^2+^) prepared complexes were studied. The outcomes revealed the following:

All complexes have a square planar geometry, while Ni(II) is square pyramidal.

Depending on the HOMO-LUMO energy gap, the DFT analysis generated that they were more stable complexes than the parent ligand.

Ni complexes are the most potent antimicrobial agent against different bacterial strains of *Escherichia coli*, *Klebsiella pneumonia*, *Staphylococcus aureus*, and *Streptococcus mutans* with inhibition zones of 65.95 ± 0.5, 57.36 ± 0.6, 69.21 ± 0.6, and 72.34 ± 0.5, respectively, which are higher than the compared reference antibiotics gentamicin (for Gram-negative) and ampicillin (for Gram-positive). Moreover, the ligand and its complexes (1S–4S) exhibit substantial anticancer effects.

The ligand and its complexes are effective against the HepG2 cell line. The IC_50_ values were ordered as follows: vinblastine (4.58) < ligand (20.45) < Ni(II) complex (41.2) < Hg(II) complex (48.5) < Cu(II) complex (182.61) < Ag(I) complex (293.95) µM. This highlights that it might be useful in cancer treatment.

Molecular docking explored the suggested interactions with the active amino acids of the ribosyltransferase and EGFR tyrosine kinase enzymes.

## Data Availability

The data presented in this study are available on request from the corresponding author.

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
