# Peer review of "Antitumor and Antibacterial Activity of Ni(II), Cu(II), Ag(I), and Hg(II) Complexes with Ligand Derived from Thiosemicarbazones: Characterization and Theoretical Studies"

_molecules, 2023, doi:10.3390/molecules28062590_

Round 1

Reviewer 1 Report (New Reviewer)

1. Delete the word “novel” from this text, could be replaced by “new”.

2. Review all graphics, subtitles are small, ariel, no pattern. This seems irrelevant but it organizes the work for the reader.

3. Table 2 has a better format, why haven't you established a standard?

4. “The presence of bands at 1634-1595 cm-1 was assigned to ν(CO3), indicating biden-94 tate carbonate groups.: what’s the matter for CO3? Also, I suggest the authors to discuss the IR part in detail, thus the refs could be cited, such as Micropor. Mesopor. Mat, 341(2022) 112098 and Inorganics, 10(2022) 202.

4. The quality of the Fig. 7 should be improved.

5. There are many Tables; I suggest them only kept within 4 Tables.

6.  There are many typesetting error, please check and correct them, such on line 158 and 330.

7. Please explain this phenomenon, “Moreover, the findings showed that the ligand and its complexes are potent against human HepG2 cell lines. Importantly, the Ni(II)complex was the most effective and exhibited concentration-dependent effects, suggesting its potential utility in cancer treatment.” Generally, Ag(I) complex may be better?

8. “Simulation of the protein-drug interaction is important in designing the structure-based drug.” Please cite related refs, such as Inorg. Chim, Acta 546(2023)121297 and Molecules, 27(2022) 7166.

Author Response

Reviewer 2 Report (Previous Reviewer 1)

Dear editor,

I notice that this manuscript has been submitted before and this is the second time for me to review it. The manuscript reported the synthesis, structures and characterizations, antitumor and antibacterial activity of Ni(II), Cu(II), Ag(I), and Hg(II) complexes with ligand derived from thiosemicarbazones. The manuscript is improved but some problems still exist.

1.     Page 4, line 131, “representing a square planar geometry for Cu2+, Ag+, and Hg2+ complexes.”, however, it is well-known that the Ag+ is ordinarily two-coordinated, why? Pleases give some detailed discussion about the geometry of the Ag+ complex; also give some references about the square-planar geometry Ag+.

2.     All the labels in Figs. 4, 5 and 8 are hardly to see, please redraw the figs and make them clear.

Round 2

Reviewer 1 Report (New Reviewer)

accept.

This manuscript is a resubmission of an earlier submission. The following is a list of the peer review reports and author responses from that submission.

Round 1

Reviewer 1 Report

Dear editor,

     The manuscript reported the synthesis, structures and characterizations, antitumor and antibacterial activity of Ni(II), Cu(II), Ag(I), and Hg(II) complexes with ligand derived from thiosemicarbazones. The manuscript can be published only after some major revisions.

1.     The English of manuscript should be improved. An example: line 64, page 2, “ligands and its were assessed……”.

2.     Page 4, line 106-107, “representing a square planar geometry for Cu2+, Ag+, and Hg2+ complexes.”, however, it is well-known that the Ag+ is ordinarily two-coordinated, why?

3.     All the labels in Figs. 4, 5 and 8 are hardly to see, please redraw the figs and make them clear.

Reviewer 2 Report

In this manuscript, Alshater et al, describe preparation of several thiosemicarbazone complexes of ni, Cu, Ag and Hg, as well as their biological properties.

Unfortunately, there are serious problems preventing me from recommendation of acceptance. Apart of the fact that the text is poorly written (there are too many grammar mistakes), characterization of the complexes is insufficient. Alone the combination of element analysis, FTIR, UV-VIS and TGA cannot reliably determine the chemical nature (structure) of the samples. PXRD is useless withous SCXRD data (those are absent). There are no mass spectra nor NMR data. Without them, it cannot be said what is the real nature of isolated substances (synthesic protocols are also poorly written - no yields, no amounts, no times!!!). This fact makes the biological studies completely irrelevant.

Unfortunately, I must recommend rejection of this manuscript.

Round 2

Reviewer 1 Report

Dear editor,

       Sorry for the late. The revised manuscript can be accepted now!

Yours sincerely,

Author Response

The manuscript has been carefully checked, the language has been edited by native speakers and all errors have been corrected

Reviewer 2 Report

Although the authors have corrected their manuscript in terms of language (using external service), the scientific problems which were highlighted on the previous stage of review remained, in fact, intact. 

The authors added two mass spectra for Ag and Ag complex, mentioning that there are peaks corresponding to the formulae. That does not explain the other numerous peaks. It can be fragmentation of this complex or some by-products or something else - this must be explained. In other words, these mass spectra do not confirm purity and nature of the whole sample (also, it remains unclear why MS was performed only for two compounds, not all four).

PXRD is NOT informative in absence of SCXRD. There are no NMR data. There is still a bit of question what are the substances which were later used for biological properties.

I still do think that this work does not fully comply with modern standards of characterization of new complexes and, thereforem this manuscript is not suitable for Molecules which is a high-IF journal.

Author Response

Responce to Reviewer 2

- Although the authors have corrected their manuscript in terms of language (using external service), the scientific problems which were highlighted on the previous stage of review remained, in fact, intact. 

Response: The manuscript has been carefully checked, the language has been edited by native speakers and all errors have been corrected.

- The authors added two mass spectra for Ag and Hg complex, mentioning that there are peaks corresponding to the formulae. That does not explain the other numerous peaks. It can be fragmentation of this complex or some by-products or something else - this must be explained. In other words, these mass spectra do not confirm purity and nature of the whole sample (also, it remains unclear why MS was performed only for two compounds, not all four).

Response: look Section 2.3(main text) and fig. S3-S7 (supplementary file)

2.3. ESI-MS spectra

The mass spectrum of the ligand exhibited a molecular ion peak at m/z = 336 amu (Calc. m/z = 334.5). The important fragment ions appear  at: m/z 77 for [C6H4-3H]+, 112 for [C6H5Cl -H]+, 127 for [C6H5NCl- 2H]+, 201 for [C7H9N3SCl]+, 228.5 for [C8H9N3SOCl-H]+, 259 for [C9H10N4OSCl-2H]+, 336 for [C15H15N4OSCl] (Fig.S3).

The mass spectra of  Ni(II), Cu(II), Ag(I) and Hg(II) complexes shows molecular ion peaks at m/z 472, 732, 618 and 717, these data are in good agreement with the proposed molecular formulas for complexes (calc. 471.54, 732.18, 618.65 and 715.35 amu), respectively. Fig. (S4 - S7).

The interpretation of the mass spectra of the complexes, Cu(II) and Ag(I) has been clarified, revealing several significant fragments. The data are in good agreement with the proposed molecular formulas for complexes, Cu(II) and Ag(I) (calc. 732.18 and 618.65 respectively), and this verifies the complexes' chemical structures.

The mass spectra of Cu(II) complex shows peak attributed to the molecular ion peak [M+] m/z at 732  (8.52%) corresponding to (C30H29Cl2CuN8O2S2). The various fragments of this complex give the peaks with different intensities at different m/z positions as at 348.91 (100.00%) (C15H15ClCuN3OS), 620.71 (11.20%) (C25H27ClCuN7O2S2), 585.03 (19.20%) (C22H22Cl2CuN6OS2), 578.58(22.59%)(C24H29ClCuN7O2S),  and 91.12 (17.94%) (C6H5N).

 The mass spectra of Ag(I) complex displays peak recognized to the molecular ion peak [M+] m/z at 618  (9.22%) corresponding to (C15H19Ag2ClN4O3S2). The various fragments of this complex give the peaks with different intensities at different m/z positions as at 127.57 (100.00%) (C6H6ClN), 600.64 (13.20%) (C15H17Ag2ClN4O2S2), 347.19 (9.20%) (C10H15AgN4OS), 90.15(14.59%)(C2H6N2S),  and 75.09 (8.94%) (C2H7N2O).

Figure S3. Mass spectrum of the ligand(H2L)

               Figure S4 Mass spectra of the Ni(II) complex (1S)

               Figure S5 Mass spectra of the Cu(II) complex (2S)

Figure S6 Mass spectra of the Ag(I) complex (3S)

               Figure S7 Mass spectra of the Hg(II) complex (4S)

- PXRD is NOT informative in absence of SCXRD. There are no NMR data. There is still a bit of question what are the substances which were later used for biological properties.

Response: Unfortunately, all our efforts to grow single crystals for all prepared complexes were unsuccessful.                                                                                                                                       

Round 3

Reviewer 2 Report

The authors added MS data, but not other methods I mentioned previously.

The work may be more or less OK for a journal with IF around 3 but NOT Molecules.

I cannot recommend acceptance.